# Cardinality Counting in "Alcatraz":
# A Privacy-aware Federated Learning Approach

## ABSTRACT

The task of cardinality counting, pivotal for data analysis, endeavors to quantify unique elements within datasets and has significant applications across various sectors like healthcare, marketing, cybersecurity, and web analytics. Current methods, categorized into deterministic and probabilistic, often fail to prioritize data privacy. Given the fragmentation of datasets across various organizations, there is an elevated risk of inadvertently disclosing sensitive information during collaborative data studies using state-of-the-art cardinality counting techniques. This study introduces an innovative privacy-centric solution for the cardinality counting dilemma, leveraging a federated learning framework. Our approach involves employing a locally differentially private data encoding for initial processing, followed by a privacy-aware federated $K$-means clustering strategy, ensuring that cardinality counting occurs across distinct datasets without necessitating data amalgamation. The efficacy of our methodology is underscored by promising results from tests on both real-world and simulated datasets, pointing towards a transformative approach to privacy-sensitive cardinality counting in contemporary data science.

## 1 INTRODUCTION

Cardinality counting problem involves determining the number of distinct elements in a set, often referred to as the "cardinality" of the set, which is of paramount importance in contemporary data analysis, supporting critical applications in various domains, such as web mining, marketing, cybersecurity, and healthcare [4, 6, 9, 21, 24, 28, 37]. In the realm of web mining, it plays a pivotal role in determining the number of unique visitors to websites, enabling businesses to optimize content, enhance user experiences, and refine marketing strategies [1, 19, 35]. In the cybersecurity domain, counting the cardinality of unique host IP addresses accessing a network resource is essential when a victim server is flooded with an enormous number of incoming malicious packets in the case of Distributed Denial of Service (DDoS) attack, thus safeguarding the integrity and availability of online services without violating honest users' privacy [23, 27, 30]. Furthermore, the problem is increasingly relevant in the smart healthcare field, where it contributes to anonymizing and preserving patient privacy in the context of large-scale medical research [20, 22]. By estimating the number of distinct patients in healthcare datasets, researchers can extract valuable insights while adhering to rigorous privacy regulations, and facilitating groundbreaking medical discoveries without compromising the confidentiality of sensitive patient information [32]. In the case of a young boy suffered with Asthma, the privacy-preserving cardinality counting technology allows the third party, for example The Department of Health, to investigate or aggregate similar cases without violating individual patient's privacy.

Cardinality counting, a critical problem in data analysis, encompasses a range of methods designed to estimate the number of unique elements within a dataset [1, 5, 14, 15, 17, 18]. These methods can be broadly categorized into deterministic and probabilistic approaches. Deterministic methods [2, 10, 25] target for exact counts of distinct elements but often face limitations due to memory constraints. On the other hand, probabilistic methods offer cardinality estimates with substantially reduced memory usage by employing statistical and hashing techniques. Prominent probabilistic methods include the HyperLogLog algorithm [16], which leverages the distribution of leading zeros in hashed values, MinHash [3], which uses random permutations to approximate cardinality, and Count-Min Sketch [11], a data structure that estimates counts of elements. These techniques have diverse applications in database management, network traffic analysis, and web analytics, addressing the need for memory-efficient cardinality estimation in large and complex datasets, ultimately contributing to more efficient and scalable data processing.

However, the existing methods do not fully consider privacy during the counting process [13]. While these techniques excel in estimating cardinality with efficiency, they often fall short when it comes to protecting sensitive information within the data from a technical standpoint. Consider the MinHash algorithm, a widely used probabilistic method for estimating cardinality by hashing elements and finding minimum hash values. In the context of a social media platform, if user profiles are represented by sets of hashed interests, MinHash can effectively estimate the number of distinct user profiles. However, if an adversary possesses knowledge of a specific user's interests and their corresponding hash values, they could query the system for the estimated cardinality of users with those same hashed interests. This query would inadvertently reveal whether that user's profile is in the dataset [31]. This technical vulnerability can pose a significant privacy risk, especially in scenarios where data security is paramount, such as protecting the anonymity of individuals in medical research datasets or safeguarding user preferences in online platforms [12]. As a result, privacy-preserving cardinality counting methods are increasingly essential to address these technical vulnerabilities and protect sensitive data [36].

**Research Gap.** The existing methods fail to provide strong privacy guarantees during the counting process. The situation where data often resides in different silos and privacy regulations such as GDPR [26] not allowing arbitrary or unrestricted data transfers raises additional challenges for cardinality counting. Consider a scenario where organizations collaborate on a data analysis project, each holding valuable but sensitive information. Traditional cardinality counting methods, although efficient, may inadvertently expose sensitive data when combined or compared across these silos. Consequently, there is a growing need for the development of a privacy-preserving cardinality counting method that can preserve

**Table 1: Summary of Notation**

| Notation | Description |
|---|---|
| $\mathcal{M}$ | The set of clients |
| $\mathcal{D}$ | Raw dataset for each client |
| $bf$ | Bloom filter |
| $bf_{\text{ref}}$ | Selected reference Bloom filter |
| $bf_{\text{dum}}$ | Generated dummy Bloom filter |
| $b$ | Each bit in a Bloom filter |
| $\mathcal{B}$ | Encoded dataset |
| $\epsilon_{LDP}$ | Privacy budget for local Differential Privacy |
| $\eta$ | Flipping probability for random response |
| $\mathcal{C}$ | The cluster centroids |
| $k_g$ | The number of clusters in $K$-means clustering |
| $k^*$ | The optimal cardinality estimation |
| $\epsilon_{\text{fed-DP}}$ | Privacy budget used in differentially privately gradient updates in federated learning |

privacy across disparate data sources, allowing organizations to collaborate on insights without compromising individual privacy.

**Contributions.** In this paper, the main contributions are as follows:

- We propose the first federated cardinality counting framework that allows cardinality counting to occur across distinct datasets without necessitating data amalgamation, prioritizing data privacy and providing a strong privacy guarantee during the counting process.
- We provide a strong privacy guarantee by utilizing data privatization with Bloom filter encoding and local Differential privacy during data encoding on the client side, and by using a federated $K$-means clustering with differentially private gradient updates.
- We conduct experimental evaluations on real and synthetic North Carolina voter registration (NCVR) datasets to validate the accuracy of the cardinality estimation with privacy-preserving federated clustering. The extensive experiments demonstrate that, even when datasets are corrupted and there are data errors and variations, high accuracy in cardinality estimation is achievable with small privacy budgets.

## 2 PRELIMINARIES

This section briefly overviews Bloom filter encoding, local differential privacy (LDP), federated learning, and $K$-means clustering with cardinality counting. To help readers better understand this paper, Table 1 lists the frequently used notations in this paper.

### 2.1 Data Privatization

In the realm of Smart Health, the collection and utilization of patient data are integral for research, diagnostics, and treatment planning. However, with the growing concerns surrounding data privacy and security, it is imperative to implement robust data privatization techniques that safeguard patient information while enabling healthcare institutions to leverage data for research and analysis. So, healthcare institutions are utilizing a data privacy technique that combines Bloom filter encoding and local differential private noise addition on the collected patient data, which includes medical records, diagnostic results, treatment histories, and more.

*2.1.1 Bloom Filter Encoding.* To mitigate privacy concerns, Bloom filter encoding is used in the first step in the data privatization. In the context of patient data, individual pieces of personal information (e.g., names and addresses) are mapped to Bloom filters. This encoding process converts sensitive data into a binary format, providing a level of anonymity.

Bloom filters are probabilistic data structures that are highly efficient for storing, processing, and computation. They are composed of bit vectors that are initially filled with zeros. To map an element $x$, $k$ independent hash functions $h_i(\cdot)$ (with $1 \le i \le k$) are used to set the corresponding bit positions in the Bloom filter $b$ to 1 (i.e., $\forall_i b[h_i(x)] = 1$). This allows for a tunable false positive rate $fpr$ so that a query returns either "definitely not" (with no error), or "probably yes" (with a probability $fpr$ of being wrong). The lower $fpr$ is, the better the utility, but the more space the filter requires. The false positive probability for encoding $n$ elements into a Bloom filter of length $\ell$ bits using $k$ hash functions is $fpr = (1 - e^{-kn/\ell})^k$, which can be adjusted by tuning the parameters $k$ and $\ell$. The main advantage of Bloom filter encoding is that it preserves the similarity/distance between records in the Bloom filter space (with a minimal utility loss) [29, 33]. For example, with string values, the $q$-grams (sub-strings of length $q$) of string values can be hashed into the Bloom filter $bf$ using $k$ independent hash functions [29], while for numerical values, the neighboring values (within a certain interval to allow fuzzy matching) of values can be hashed into the Bloom filter [33].

*2.1.2 Locally Differentially Private Data Encoding.* To further enhance privacy, local differential private noise is added to each encoded record. This technique ensures that even with access to the encoded data, it remains challenging to infer specific details about individual patients. Local Differential Privacy (LDP) mechanisms, similar to Randomized Aggregatable Privacy-Preserving Ordinal Response (RAPPOR), are employed to inject noise into the encoded data. LDP guarantees that the privacy of each patient's information is preserved while enabling aggregate-level analysis. This technique provides $\epsilon$-local Differential Privacy guarantees by randomly flipping each bit in the Bloom filter of encoded records locally by the data providers with a probability of $\eta = \frac{1}{1+e^{\epsilon_{LDP}}}$. To ensure privacy, the randomized response technique is employed to alter the Bloom filters, providing $\epsilon$-local Differential Privacy guarantees. This is done by randomly flipping each bit in the Bloom filter of encoded records locally by the data providers with a probability of $\eta = \frac{1}{1+e^{\epsilon_{LDP}}}$.

**DEFINITION 1 (ADJACENT BLOOM FILTERS).** *Adjacent Bloom filters are two Bloom filters $bf$ and $bf'$ of length $\ell$ bits that differ by only one-bit position, i.e., $\forall_{i,1 \le i \le \ell \text{ and } i \ne j} b_i = b_i'$ and $b_j \ne b_j'$.*

**LEMMA 1 ($\epsilon$-LDP FOR BLOOM FILTERS).** *Flipping the bits in Bloom filters with $\frac{1}{1+e^{\epsilon_{LDP}}}$ probability makes the bits in the Bloom filters $\epsilon$-local differentially private.*

The algorithm used for the two steps data privitization is shown in Algorithm 1. By utilizing LDP on the bits of Bloom filters, we can make them resistant to cryptanalysis attacks that target sensitive bits [7]. So, with the incorporation of Bloom filter encoding and local differential private noise addition, the healthcare institutions

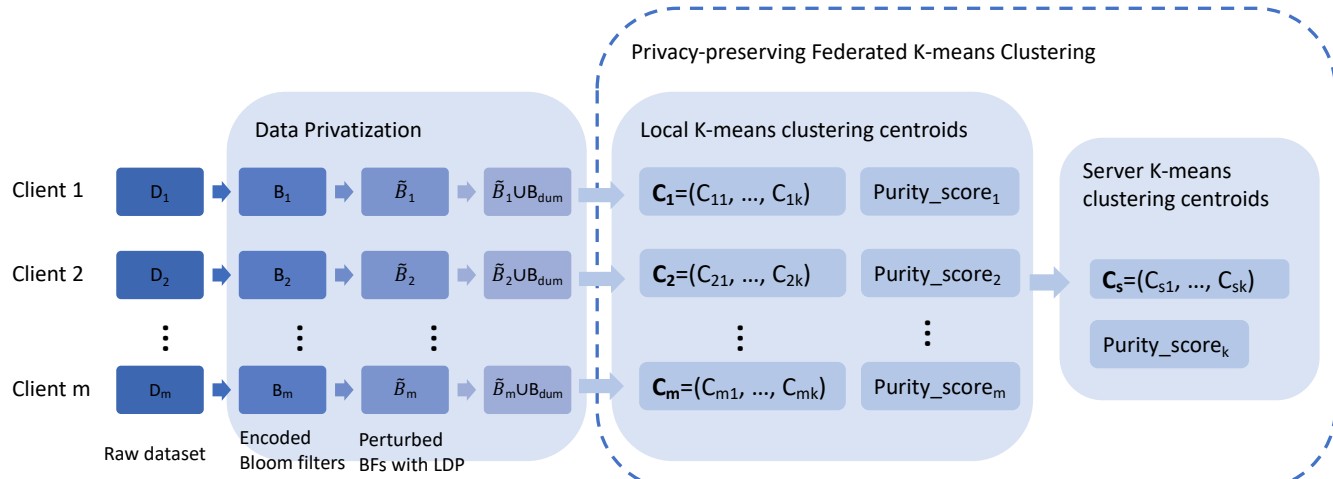

**Figure 1: An outline of our system model for privacy-preserving cardinality estimation with federated $K$-means clustering.**

---

**Algorithm 1** *Data Privatization for one client with Locally Differentially Private Data Encoding*

---
1: **Inputs:**
    Raw dataset from a client: $D$,
    Privacy budget: $\epsilon_{LDP}$
2: **Outputs:**
    Encoded dataset: $\mathcal{B}$
3: **Initialize:**
    $\mathcal{B} \leftarrow \Phi$
4: **for** each record $x \in D$ **do**          ▷ Do for each record in raw dataset
5:     $bf_{step1} = \text{BloomFilterEncode}(x)$,          ▷ First step: Bloom filter encoding
6:     $bf_{step2} \leftarrow bf_{step1}$          ▷ Second step: add Local DP noise
7:     **for** each bit $b$ in Bloom filter $bf_{step2}$ **do**          ▷ For each bit in the Bloom filter
8:         $\eta = \frac{1}{1+e^{\epsilon_{LDP}}}$
9:         $p = random[0,1]$          ▷ Randomly generate a number between 0 and 1
10:        **if** $p \geq \eta$ **then**          ▷ flip each bit $b$ in the Bloom filter with flipping probability $\eta$
11:            $b = b \oplus 1$
12:        **end if**
13:    **end for**
14:    $\mathcal{B} = \mathcal{B} \cup bf_{step2}$
15: **end for**

---

achieve a level of data privatization that safeguards patient confidentiality. This privatized data can now be utilized for various purposes, including medical research, data analysis, and treatment optimization, without compromising patient privacy. Researchers and healthcare professionals can obtain valuable insights from the data while complying with strict privacy regulations.

## 2.2 $K$-means Clustering for Cardinality Counting

Cardinality counting enables healthcare providers and administrators to comprehensively understand their patient populations, such as patients with specific medical conditions, age groups, or geographic locations. Accurately estimating the number of patients helps to detect early diseases, track the spread of diseases, and evaluate the success of interventions. $K$-means clustering is a powerful tool for determining the cardinality of health-related entities. This

technique groups similar health data points or entities into clusters based on their shared characteristics, and the optimal number of clusters implies the number of unique individuals across multiple datasets from different sources. However, with the absence of labeled data, it is difficult to find the optimal $k^*$ by using the traditional Elbow method. Previous studies have mainly concentrated on centralized approaches, where a clustering algorithm is employed at the linkage unit. In the previous work by [36], Differentially Private Bloom filters were sent to the linkage unit for the $K$-means clustering. At the linkage unit, the dataset from multiple data providers is integrated first, and additional reference and dummy Bloom filters are added to the integrated dataset to help evaluate the purity and completeness of each cluster, so as to help in determining the optimal $k^*$ in the $K$-means clustering. For a reference Bloom filter, the dummy Bloom filters are generated by randomly flipping each bit in the Bloom filters with a flipping probability $\eta$.

## 3 PROPOSED METHODOLOGY

In this section, we first describe the centralized $K$-means method for precise cardinality estimation. Then, we extend the centralized $K$-means method to federated $K$-means with privacy guarantees during the federated learning process.

### 3.1 System Model

The system model of our proposed method for privacy-preserving cardinality counting is illustrated in Fig. 1. There is a fixed set of $m$ clients $\mathcal{M}$, each client $i \in \{1, \cdots, m\}$ with a fixed local data set $D_i$ of size data $n_i$, who want to cooperate with each other to train a $K$-means clustering machine learning model. At client $i$, the personal identifying information (PII) in records $r_i$ is initially encoded into Bloom filters $bf_i$ and then perturbed with LDP as $bf_i'$. One record is used to generate one Bloom filter. In the previous work, the encoded and perturbed records from multiple different data owners $B_i, \cdots, B_m$ are sent to a linkage unit, which, to a certain extent, could compromise privacy by potentially deducing the encoded data.

**Algorithm 2** *Privacy-preserving Federated K-means Clustering*

---

1: **Inputs:**
    $k_g$, $B_i$, $i \in \mathcal{M}$
2: **Outputs:**
    $k^*$
3: **for** $k = k_{min}, \cdots, k_{max}$ **do**
4:    $k_g = k$
      **Initialization:**
5:    **for** each client $i \in \mathcal{M}$ **do**
6:      $S_{i,r=0}, C_{i,r=0} = \text{kmeans++init}(B_i, k_g)$    ▷ Get cluster means using kmeans++ initialization
7:    **end for**
8:    $S_{s,r=0} = [S_{1,r=0}|S_{2,r=0} \cdots S_{m,r=0}]$
9:    $C_{s,r=0} = \frac{1}{m} \sum_{i=1}^{m} C_{i,r=0}$    ▷ Get global initial cluster centroids
10:    **for** each round $r = 1, \cdots, r_{max}$ **do**
      **Server Executes:**
11:      Select a subset of $\mathcal{M}' \in \mathcal{M}$ clients
12:      Send $C_{s,r}$ to $\mathcal{M}'$ clients
13:      Recieve $S_{i,r+1}, C_{i,r+1}, \mathcal{P}_{i,r+1}$ from clients $i \in \mathcal{M}'$
14:      $S_{s,agg} = [S_1|S_2 \cdots S_{m'}]$
15:      $C_{s,agg} = \frac{1}{m'} \sum_{i=1}^{m'} C_{i,r+1}$    ▷ Aggregate centroids from clients
16:      $\mathcal{P}_{s,r+1} = \frac{\sum_{i=1}^{m'} n_{\text{ref},i} \mathcal{P}_i}{\sum_{i=1}^{m'} n_{\text{ref},i}}$    ▷ Aggregate purity scores
17:      $S_{s,r+1}, C_{s,r+1} \leftarrow \text{reallocate}(S_{s,agg}, C_{s,agg})$    ▷ Reallocate empty clusters by randomly assigning low score points to form new clusters to maintain $k_g$ non-empty clusters
      **Client Executes:**
18:      **for** each client $i \in \mathcal{M}'$ **do**
19:        **while** $I_i \leq I_{loc}$ **do**
20:          $S_{i,r+1}, C_i \leftarrow \text{kmeans}(B_i, k_i, C_{s,r})$ ▷ Update locally cluster means for each client using kmeans
21:        **end while**
22:        $\mathcal{P}_{i,r+1} \leftarrow \text{purity\_score}(B_{\text{ref},i}, B_{\text{dum},i}, S_{i,r+1}, C_i)$▷ Obtain purity score by Equation (4)
23:        $C_{i,r+1} = C_i + \mathcal{N}_i(0, \sigma_{\text{DP}}^2)$    ▷ Obtain DP-perturbed cluster means
24:        Send $S_{i,r+1}, C_{i,r+1}, \mathcal{P}_{i,r+1}$ to the server
25:      **end for**
26:    **end for**
27:    $\mathcal{P}_k = \mathcal{P}_{s,r_{max}}$
28: **end for**
29: $k^* = \arg\max_{k \in [k_{min}, \cdots, k_{max}]} \mathcal{P}_k$

---

We proposed a federated $K$-means clustering approach that does not require sharing encoded records or dummy Bloom filters. Instead, a central server computes the centroids of the clusters by receiving centroids gradient updates from the participating clients. Then, the similar Bloom filters corresponding to the same individual/patient are grouped into one cluster. The clustering centroids are determined by the server. For details, our proposed method stores the perturbed bloom filters locally. And each client $i$ computes the local $K$-means clustering centroids $C_i, = (C_{i1}, \cdots, C_{ik}), i \in \{1, \cdots, m\}$ when cluster number is $k$. Then, the client only sends the centroids updates gradient to the server to preserve data privacy. The server aggregates the received centroids gradients $C_1, \cdots, C_m$ from the $m$ clients and computes the sever centroids $C_s = (C_{s1}, \cdots, C_{sk})$. The server is responsible for determining the optimal number of clustering centroids by checking the clustering performance for each $k \in [k_{min}, k_{max}]$. The optimal number $k^*$ is estimated as the cardinality of records from multiple databases and reports.

## 3.2 Federated $K$-means Clustering Problem

The federated $K$-means clustering problem attempts to group similar Bloom filters into $K$ groups of equal variance across different clients by minimizing a criterion known as the inertia or within-cluster sum-of-squares. The means are commonly referred to as

the cluster "centroids" $C = C_1, \cdots, C_K$. Given a set of Bloom filters $\mathcal{B} = \{\mathcal{B}_1, \cdots, \mathcal{B}_m\}$ distributed over $m$ clients, where each client $i$ owns a Bloom filter $bf_i \in \mathcal{B}_i$, where $bf_i$ is a $\ell$ dimension vector, the federated $K$-means clustering aims to partition the $|\mathcal{B}|$ Bloom filters owned by $m$ clients into $K (\leq |\cup_{i \in \mathcal{M}} \mathcal{B}_i|)$ sets $\mathcal{S} = S_1, \cdots, S_K$ so as to minimize the within-cluster sum of squares. We assume that $m'$ clients are randomly selected from $\mathcal{M}$ for the federated $K$-means clustering for each round, and denote the selected client set as $\mathcal{M}'$. The federated $K$-means clustering problem is formulated by minimizing the within-cluster sum of squares averaged over all clients, as given by

$$\arg\min_{\mathcal{S}, \mathbf{W}} \frac{1}{m'} \sum_{i=1}^{m'} \sum_{j=1}^{K} w_{i,j} \sum_{bf \in S_j} \|bf - C_j\|^2, \quad (1)$$

where $C_j \in C$ is the centroid in cluster $S_j$; and $w_{i,j}$ is weight vector of client $i$ for the $j^{th}$ centroid.

## 3.3 Threat Model

By combining Bloom filter encoding and local differential privacy with random response, the data providers utilize a two-step data privatization process that offers two-layer privacy guarantees. The initial layer provides privacy assurances due to the fact that different elements being mapped to the same bits in the Bloom filters can cause collisions, thus introducing uncertainty when decoding. The second layer of privacy with LDP, offers a verifiable assurance of privacy against cryptanalysis attacks as discussed in [7, 8].

Federated $K$-means clustering groups similar Bloom filters into the same cluster across different clients without the need to share the raw data to the centralized server, thus reducing the risks of raw data leakage. However, it still faces potential threat models that can compromise the privacy and security of participants, such as membership inference attacks, model evasion attacks, and data poisoning. Differential privacy is a technique that can help mitigate these threats. Implementing differential privacy (DP) in federated learning requires a delicate balance between privacy and utility. Too much noise can be detrimental to the learning process and thus the estimation of cardinality.

## 3.4 Privacy-preserving Federated $K$-means Clustering for Cardinality Counting

In this subsection, we propose a privacy-preserving federated $K$-means clustering algorithm for cardinality counting.

*3.4.1 Privacy-preserving Federated K-means Clustering.* Privacy-preserving federated $K$-means clustering is a modern approach to machine learning that allows data to be processed locally instead of being sent to an external aggregator. This makes it more secure for multiple entities, such as hospitals, diagnostic centers, clinics, and laboratories, to work together to train machine learning models without having to share their raw data with a third party.

In our proposed algorithm, each client $i \in \mathcal{M}$ first computes the initial centroids $C_{i,r=0}$ kmeans++ with their encoded dataset $\mathcal{B}_i$, then sends them to the server to aggregate the global initial cluster centroids $C_{s,r=0}$. Then, the server randomly selects a subset of clients $\mathcal{M}'$ for local centroids aggregation and broadcasts the aggregated global centroids to all clients. Each client uses the global

centroids received from the server and updates the cluster centroids using the local dataset. After finishing the local updates at each selected client, a DP noise is added to the local obtained cluster centroids. They then send the perturbed cluster centroids $C_{i,r+1}$ to the server for aggregation. The server collects the new cluster centroids that have been sent by the chosen clients. In the final aggregation round, the clients report the completeness and accuracy of the clusters by assessing the clustering status of the reference and dummy Bloom filters. The server is able to calculate a total purity score, which provides an understanding of the completeness and purity of the global centroids clusters across all clients. In what follows, we provide more details on the calculations of the DP noise and the purity score.

*3.4.2 Differential Privacy.* Consider at most $T$ exposures of the local models of the clients during the federated clustering process. To guarantee the $(\epsilon, \delta)$-DP demand of the individual dataset under $T$ global aggregations, the standard deviation of the DP noises added by the clients is given by [34]

$$\sigma_{\text{DP}} = \frac{T\Delta s}{\epsilon} \sqrt{2 \ln\left(\frac{1.25}{\delta}\right)}. \quad (2)$$

A bigger $\sigma_{\text{DP}}$ leads to a smaller $\epsilon$, i.e., greater privacy protection, while a bigger $T$ incurs a higher chance of privacy leakage. In addition, the sensitivity $\Delta s$ depends on the sizes of local datasets. An increase in the size of the data sets can lead to a decrease in $\Delta s$ and, in turn, a decrease in $\sigma_{\text{DP}}$.

*3.4.3 Purity Score Calculation.* In this research, we modify the technique from [36] and expand it to suit the federated learning between multiple clients and a server to determine the ideal $k$ value for unsupervised $k$-means clustering. A set of reference Bloom filters with known training labels is generated to evaluate the clustering performance. For each client $i$, a subset of $n_{\text{ref},i}$ numbers of Bloom filters is selected as reference Bloom filters set $B_{\text{ref},i}$. Then, for each reference Bloom filter $bf_{\text{ref},i,j} \in B_{\text{ref},i}$, a set of $n_{\text{ref},i,j}$ numbers of corresponding dummy Bloom filters $B_{\text{dum},i,j}$ are generated.

The Euclidean distance between a reference Bloom filter $bf_{\text{ref}}$ and its corresponding dummy Bloom filter $bf_{\text{dum}}$ is:

$$\|bf_{\text{ref}}, bf_{\text{dum}}\|_2 = \sqrt{\sum_{i=1}^{\ell}(b_{\text{ref},i} - b_{\text{dum},i})^2},$$

where $\ell$ is the length of the Bloom filter, $b_{\text{ref},i}$ is the $i^{th}$ bit in the reference Bloom filter $bf_{\text{ref}}$, and $b_{\text{dum},i}$ is the $i^{th}$ bit in the dummy Bloom filter $bf_{\text{dum}}$. Assuming that the Euclidean distance $\|bf_{\text{ref}}, bf_{\text{dum}}\|_2$ is less than a constant integer value (threshold) $r \in [0, \ell]$, then the original Bloom filter and the perturbed Bloom filter are grouped into the same cluster. The probability of $bf_{\text{ref}}$ and $bf_{\text{dum}}$ being classified as the same entity is:

$$P(\|bf_{\text{ref}}, bf_{\text{dum}}\|_2 \leq r) = \frac{1}{2} + \frac{1}{2} \text{erf}\left(\frac{r^2 - \ell\eta}{\sqrt{2\ell\eta(1-\eta)}}\right). \quad (3)$$

The detailed proof is provided in [36].

Due to the feature of federated learning that all data are stored and processed locally on the client side, the reference Bloom filters are randomly selected from the encoded dataset. The corresponding

dummy Bloom filters are generated by randomly flipping each bit in the reference Bloom filter with the flipping probability of $\eta$.

In a similar fashion to the research conducted in [36], this work utilizes reference and dummy Bloom filters to assess the performance of $K$-means clustering and to determine the optimal number of clusters $k^*$ in unsupervised clustering techniques when labeled data is scarce. In this study, the reference and dummy Bloom filters are stored locally on the client side, and only the size of the reference Bloom filters $n_{\text{ref}}$ is shared with the server.

Following each $K$-means clustering process with $k_g$, the Bloom filter in client $i$ is assigned to a cluster with a centroid $c$ from the set $C_S$. The purity function for a reference Bloom filter $bf_{\text{ref},i,j}$ clustered in the cluster with the centroid $c \in C_s$ is defined as: $\mathcal{P}(bf_{\text{ref},i,j}) = \frac{n_{j,\text{dum},c}}{n_{j,\text{dum}}+n_c-n_{j,\text{dum},c}}$, where $n_{j,\text{dum},c}$ is the number of dummy records for the reference Bloom filter $bf_{\text{ref},i,j}$ of client $i$ that are grouped in the same cluster with label $c$, $n_c$ is the number of Bloom filters that are grouped into the cluster with label $c$, $n_{j,\text{dum}}$ is the total number of dummy records for the reference Bloom filter $bf_{\text{ref},i,j}$. Then, the purity for client $i$ is given by:

$$\mathcal{P}_i = \sum_{bf_{\text{ref},i,j} \in B_{\text{ref},i}} \mathcal{P}(bf_{\text{ref},i}). \quad (4)$$

With this purity function, each client $i$ measures the purity and the completeness of generated server cluster centroids by checking whether each of their reference Bloom filters is grouped with the corresponding dummy Bloom filters and sends the purity $\mathcal{P}_i$ to server. The server determines the overall purity at $k_g$ clusters in $K$-means clustering from all clients by:

$$\mathcal{P}_{k_g} = \frac{\sum_{bf_{\text{ref},i,j} \in B_{\text{ref},i}} n_{\text{ref},i}\mathcal{P}_i}{\sum_{i \in \mathcal{M}} n_{\text{ref},i}}. \quad (5)$$

The server then determines the optimal number of clustering centroids by checking the overall purity score $\mathcal{P}_{k_g}$ for each $k_g \in [k_{min}, k_{max}]$, and therefore the cardinality of the records of multiple clients is given by:

$$k^* = \underset{k_g \in [k_{min}, \cdots, k_{max}]}{\arg\max} \mathcal{P}_{k_g}. \quad (6)$$

The detailed steps of the privacy-preserving federated $K$-means clustering for cardinality estimation are shown in Algorithm 2. This algorithm outlines the complete process.

## 4 EXPERIMENTAL EVALUATION

**Dataset.** We conducted our experiments using data taken from the North Carolina Voter Registration (NCVR) database[1] to simulate patient records stored in various healthcare facilities. This database contains records of voters in the North Carolina State, USA. Ground-truth is available based on the voter registration identifiers to evaluate the accuracy of our proposed cardinality estimator in our experiments. Note that the ground truth is not always available in real applications. We used given name (string), surname (string), suburb (string), postcode (string), and gender (categorical) attributes as PII for the linkage. We obtained two collections of data from the dataset, the clean datasets and the corrupted datasets, each with 10 datasets for 10 clients. The ground truth cardinality for all the ten datasets is 360, meaning that the datasets contained records

---

[1]Available from http://dl.ncsbe.gov/data/

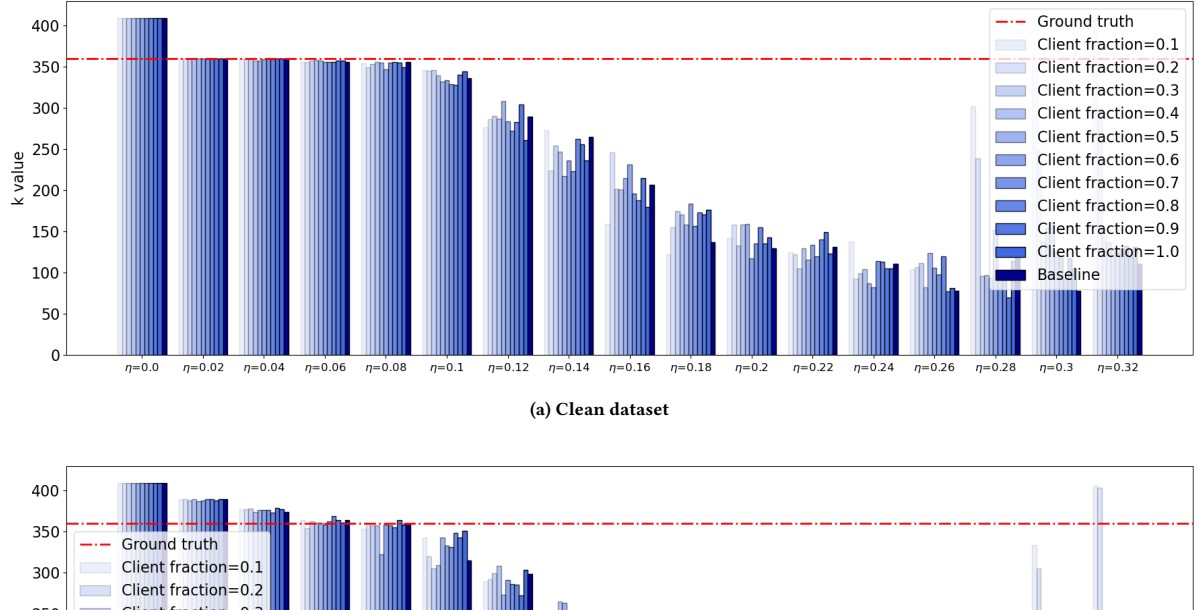

(a) Clean dataset

(b) Corrupted dataset

**Figure 2: Estimated cardinality ($k^*$) with federated clustering with different client sample fractions** $[0.1, 0.2, 0.3, 0.4, 0.5, 0.6, 0.7, 0.8, 0.9, 1.0]$ **versus flipping probabilities compared with the baseline method. In these experiments, there is no noise added in data encoding or gradient updates in federated clustering.**

for 360 distinct voters/patients. The first set of datasets consists of duplicates of the same person with no changes or corrupted PII values, while the second set of datasets includes duplicates with altered or corrupted PII values (20% of records) to imitate real-world data mistakes and discrepancies. We implemented the prototype of our proposed algorithm in Python 3.9.4, and ran all experiments on an AMD EPYC 7543 32-Core Processor. The authors provide access to the programs and test datasets.

**Baseline Method.** We compare our federated clustering results with the central clustering approach. In the central clustering approach, the clients sent their locally differentially private encoded data to the server for aggregation, followed by $K$-means clustering.

**Parameter Setting.** We assume that there are 10 clients participating in cardinality counting with a total ground truth of 360. The $q$-gram length is 2 in Bloom filter encoding. The Bloom filter length is 500. The false positive probability is extremely low, with a value of $fpr = 6.5 \times 10^{-6}$ in the current Bloom filter encoding settings. Privacy budgets used for local DP are $\epsilon_{\text{LDP}} = [1.0, 2.0, 3.0, 4.0, 5.0, 10.0, 1000.0]$, where $\epsilon = 1000.0$ implies no local DP noise is added in data encoding. The default reference Bloom

filters pick ratio used is 0.1, the default dummy Bloom filter number for each reference Bloom filter is set to be a uniformly distributed randomized number between $[1, 10]$, and the flipping probability for the dummy/noisy Bloom filters is used in the range $[0.0 - 0.32]$, with a step of 0.02. For the federated clustering algorithm, the fractions of clients sampled per round are varied in the range $[0.1 - 1.0]$ with a step size of 0.1. We vary the flipping probabilities in the dummy Bloom filters and evaluate the $k$ value as it impacts the clustering quality according to the data quality and privacy budget. We use different values of the fractions of clients sampled per round, local differential privacy values and federated clustering privacy budget to check the cardinality counting accuracy.

**Discussion.** We first compare the optimal $k^*$ value provided by our algorithm with the ground-truth cardinality with different flipping probabilities used in the clustering algorithm with different values of the fractions of clients sampled per round without any noise added in either Bloom filter data encoding or gradient updates in the federated clustering algorithm. The range of clients sampled for federated clustering is in the range $[0.1, 1.0]$ with a step size of 0.1. The central $K$-means clustering is the benchmark and shown as the

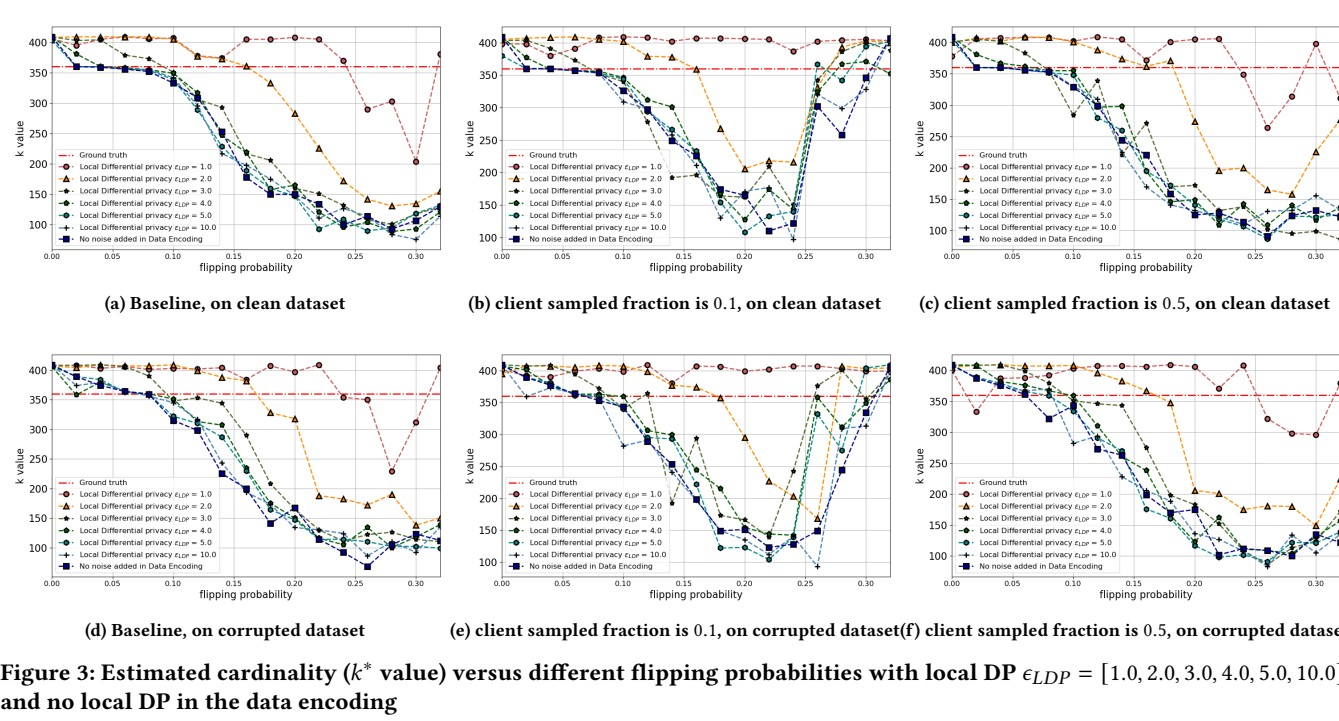

(a) Baseline, on clean dataset

(b) client sampled fraction is 0.1, on clean dataset

(c) client sampled fraction is 0.5, on clean dataset

(d) Baseline, on corrupted dataset

(e) client sampled fraction is 0.1, on corrupted dataset (f) client sampled fraction is 0.5, on corrupted dataset

**Figure 3: Estimated cardinality ($k^*$ value) versus different flipping probabilities with local DP $\epsilon_{LDP} = [1.0, 2.0, 3.0, 4.0, 5.0, 10.0]$ and no local DP in the data encoding**

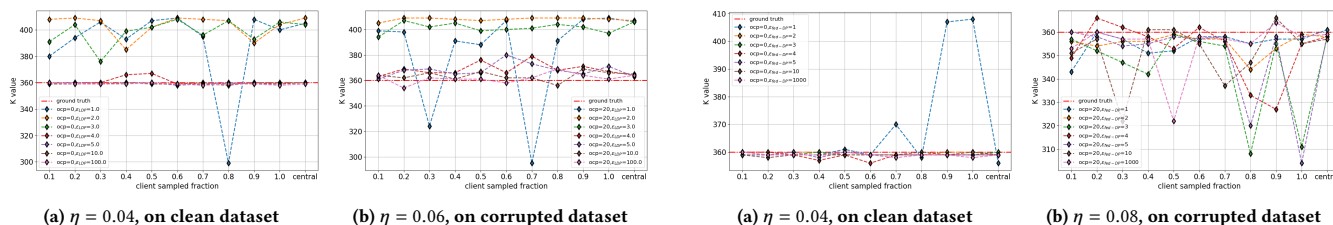

(a) $\eta = 0.04$, on clean dataset

(b) $\eta = 0.06$, on corrupted dataset

(a) $\eta = 0.04$, on clean dataset

(b) $\eta = 0.08$, on corrupted dataset

**Figure 4: Estimated cardinality ($k^*$ value) versus clients per round fractions with different Local DP $\epsilon_{LDP} = [1.0, 2.0, 3.0, 4.0, 5.0, 10.0, 100.0]$**

**Figure 5: Estimated cardinality ($k^*$ value) versus clients per round fractions with different federated privacy budget $\epsilon_{LDP} = [1.0, 2.0, 3.0, 4.0, 5.0, 10.0, 1000.0]$**

baseline in the plots. The experiments are on both clean datasets and corrupted datasets. As shown in Figure 2a, without noise added to either the data encoding side on the clean datasets, or the gradient updates in the learning process, the estimated cardinalities by the baseline and federated clustering are very close to the ground truth when the flipping probability ranges from approximately 0.02 to 0.08. For the corrupted datasets as shown in Figure 2b, the estimated cardinalities by all methods are close to the ground truth when the flipping probability ranges from approximately 0.06 to 0.08. When the dataset contains errors, typos and mistakes, a higher flipping probability is needed to get accurate estimations of the cardinality. The optimal values of $k^*$ found by federated $K$-means clustering and central clustering method are almost the same with trivial differences. When the probability of flipping is higher than 0.28, the cardinality estimation by federated $K$-means clustering with a small fraction of clients sampled per round (e.g., 0.1, 0.2, and 0.3) tends to converge towards the ground truth value.

We further evaluate how the cardinality estimation performance is affected by the local DP $\epsilon_{LDP}$ in data privatization of the clients. We compare the results between federated clustering and the Baseline method for different $\epsilon_{LDP}$ values in the range $[1.0, 10.0]$ with a step size 0.1 as well as no local DP added with different flipping probabilities, for both the clean and corrupted datasets. As illustrated in Figure 3, when the privacy budget is greater than or equal to 4.0, the cardinality estimates from the federated clustering are identical to the ground truth. For the clean datasets, when local DP $\epsilon_{LDP}$ is no less than 3.0, the optimal $k^*$ can be achieved with flipping probability $\eta$ in range $[0.02, 0.08]$ as shown in Figure 3b, and Figure 3c. Note that with small local DP $\epsilon_{LDP} = 3.0$, 24 out of 500 bits have been flipped, which implies strong privacy guarantees provided by Data privatization for the clients. Compared to the baseline central $K$-means clustering as shown in Figure 3a, federated clustering achieves good cardinality estimation with small clients sampled per round fraction 0.1 and 0.5 in Figure 3b, and Figure 3c, which means only one or five clients have been selected

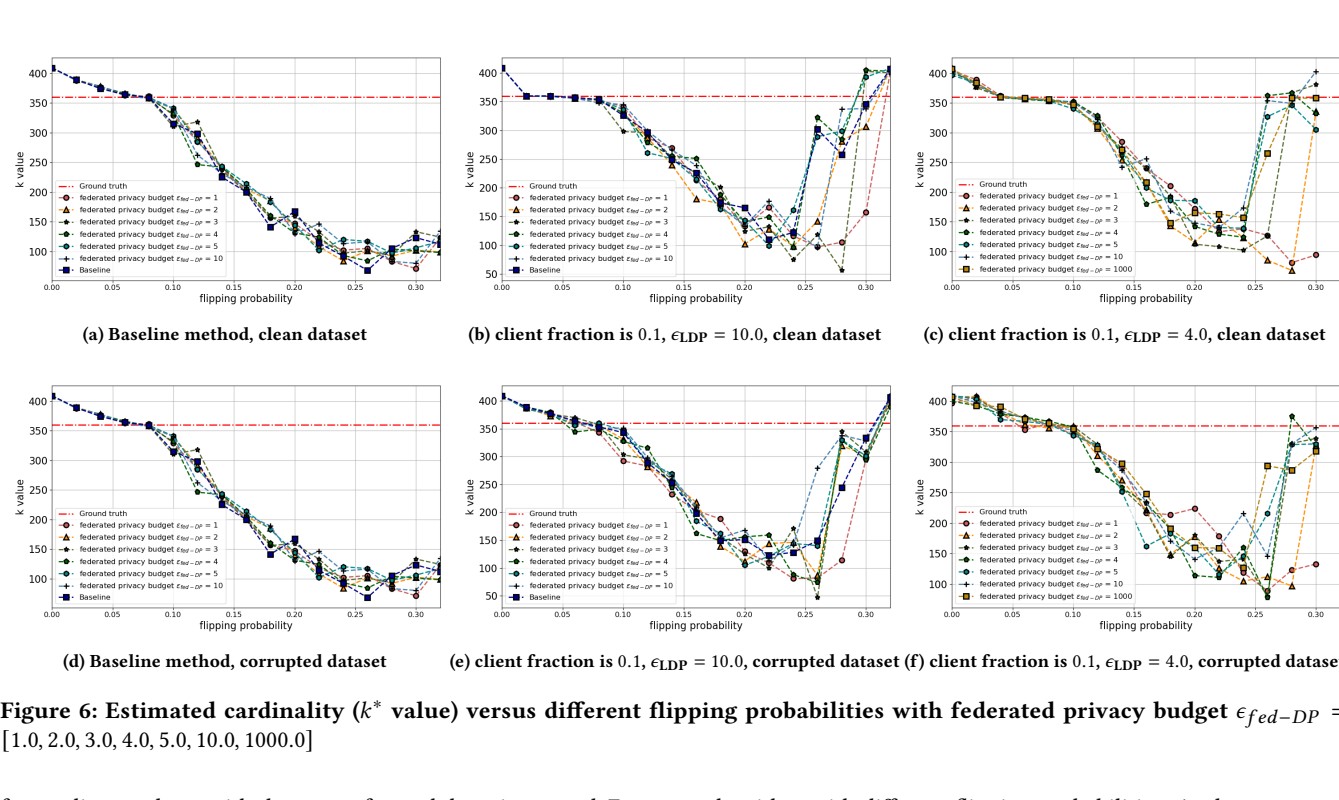

**(a) Baseline method, clean dataset**

**(b) client fraction is** $0.1$, $\epsilon_{\mathrm{LDP}} = 10.0$, **clean dataset**

**(c) client fraction is** $0.1$, $\epsilon_{\mathrm{LDP}} = 4.0$, **clean dataset**

**(d) Baseline method, corrupted dataset**

**(e) client fraction is** $0.1$, $\epsilon_{\mathrm{LDP}} = 10.0$, **corrupted dataset** **(f) client fraction is** $0.1$, $\epsilon_{\mathrm{LDP}} = 4.0$, **corrupted dataset**

**Figure 6: Estimated cardinality ($k^*$ value) versus different flipping probabilities with federated privacy budget $\epsilon_{fed-DP}$ =** $[1.0, 2.0, 3.0, 4.0, 5.0, 10.0, 1000.0]$

for gradient updates with the server for each learning round. For corrupted datasets, the cardinality estimation performances are similar to clean datasets. As shown in Figure 3e, Figure 3f and Figure 3d, the privacy-preserving federated $K$-means clustering with LDP greater than 4.0 can achieve good cardinality estimates as the baseline central clustering method when the flipping probability is around 0.06 to 0.08, which is slightly narrower than the optimal flipping probability range found on clean datasets.

We evaluate the cardinality estimations resulting from the varying fractions of clients sampled per round by adjusting the flipping probability value. The results are shown in Figure 4 and Figure 5. The results of federated clustering and central clustering for both the clean and corrupted datasets are similar, indicating that updating the gradient of the server with only a small number of clients in each round does not significantly complicate the determination of the optimal $k^*$ in $K$-means clustering. Similarly, in Figures 4a and 4b, a smaller flipping probability is required to identify the optimal $k^*$ for the corrupted datasets ($\eta = 0.04$) compared to the clean datasets ($\eta = 0.06$). Similarly, the flipping probability for optimal $k^*$ on the corrupted datasets is higher than that on the clean datasets, as demonstrated in Figures 5a and 5b. The former is 0.08, while the latter is 0.04. The average Euclidean distance between the reference Bloom filter and its dummy Bloom filter is greater in corrupted datasets with data errors and variance than in clean datasets with no data errors.

We evaluate the impact of the values of federated privacy budgets $\epsilon_{\text{fed-DP}}$ on the accuracy of cardinality estimation. We compare the results of federated clustering and the Baseline approach for various $\epsilon_{\text{fed-DP}}$ values in the range of $[1.0, 2.0, 3.0, 4.0, 5.0, 10.0]$ and no DP noise added to the gradient updates in the federated clustering

algorithm with different flipping probabilities. As demonstrated in Figure 6, when the flipping probability is between 0.02 and 0.1, the cardinality estimations are highly accurate for clean datasets, and when the flipping probability is between 0.06 and 0.1, the cardinality estimations are highly accurate for corrupted datasets. When $\epsilon_{\mathrm{LDP}}$ is set to 4.0, as illustrated in Figure 6c, the task of estimating cardinality is not more difficult than when $\epsilon_{\mathrm{LDP}}$ is set to 10.0, as demonstrated in Figure 6b.

## 5 CONCLUSION

In this paper we propose a privacy-preserving federated clustering approach for cardinality counting. The proposed methodology leverages Bloom filter encoding and local differential privacy to ensure data privacy while enabling healthcare institutions to leverage data for research and analysis. The approach utilizes a federated learning framework that allows cardinality counting to occur across distinct datasets without necessitating data amalgamation, prioritizing data privacy. The proposed methodology addresses the privacy challenges and enables secure, real-world applications of cardinality counting in data analytics, providing a strong privacy guarantee during the counting process. We conducted experimental evaluation on real and synthetic North Carolina voter registration (NCVR) datasets to verify its accuracy and show its resilience to data errors when estimating cardinality across multiple data providers while preserving individual privacy. Accurate cardinality estimation can be achieved even when a small portion of clients (10%) take part in each learning round, with a low level of data privacy and a small federated privacy budget. One future work is to use federated $K$-means clustering with a greater number of $K$ and further improve the complexity of the implementation.

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
