# OpenReview forum: "Cardinality Counting in "Alcatraz": A Privacy-aware Federated Learning Approach"
_ACM.org/TheWebConf/2024/Conference — TheWebConf24_

### Official Review · Reviewer_cwZd · 2023-11-19

**Novelty:** 6
**Technical Quality:** 5

**Review:**

Paper Summary:

This paper proposes a privacy-preserving cardinality counting algorithm. Specifically, the algorithm combines Bloom Filter Encoding, Locally Differentially Private Data Encoding, and a DP-perturbed federated K-Means. Cardinality counting has significant applications across various domains such as web mining, marketing, cybersecurity, and healthcare. However, the existing methods do not fully consider privacy during the counting process. The study proposes a privacy-centric solution for cardinality counting that can reduce privacy risks.

Strengths:

- The research problem in this paper is well-motivated.

- The paper is well-written and easy to follow.


Weakness:

- Some technique details need to be clarified.

- Lack of practical guidance on selecting the hyperparameters.

This paper studies an important problem in contemporary data analysis, the authors provide sufficient background knowledge and motivation examples to formulate the problem. The paper is well-written and easy to follow. The experiments are performed on a real-world dataset. While I enjoyed reading this paper very much, I have some concerns about the technique details.

Firstly, the main contribution of this paper is a modified version of previous work [34,36]. The authors design a purity score calculation part for the privacy-preserving federated K-means clustering. When the selected clients share the updated centroids with the server, whether this is guaranteed privacy-preserving is not proven.

Secondly, there are many hyperparameters that can affect the trade-off between estimation accuracy and privacy protection ability. The authors need to provide some practical guidance on selecting the hyperparameters. For example, the authors can provide some empirical results on how the number of clusters affects the model's utility and efficiency.

Thirdly, the authors do not analyze the computation complexity of the proposed algorithm. The authors need to provide some theoretical analysis of the computation complexity.

Fourth, the authors do not provide a convergence guarantee. In the proposed system, it is unclear whether the server needs to pre-decide a K for each client. What if the selected clients hold different Ks? Can the server side still converge to perform the clustering?

Besides, it would be great if the authors could provide some empirical results on the scalability of the proposed algorithm. For example, the authors can provide some empirical results on how the number of clients affects the model's utility and efficiency.

Minor comments:
- In Section 2.2, a ''linkage unit'' is mentioned to collect data from multiple clients. What is a linkage unit? How does it differ from a server?
- Citations 14 and 15 seem repeated.

**Questions:**

1. How to determine the k for the K-means clustering in the hetogeneous FL setting?
2. Whether the proposed algorithm can be applied to the case where the number of clusters is different for each client?
3. Are there any potential privacy risks in the proposed algorithm?
4. How to guarantee the convergence of the proposed algorithm?
5. Is there any theoretical analysis of the computation complexity?

**Ethics Review Description:**

I do not see any ethical issues with this paper.

**Reviewer Confidence:**

3: The reviewer is confident but not certain that the evaluation is correct

**Scope:**

4: The work is relevant to the Web and to the track, and is of broad interest to the community

---

### Official Review · Reviewer_8Vhp · 2023-11-23

**Novelty:** 7
**Technical Quality:** 6

**Review:**

This study introduces an innovative privacy-centric solution for the cardinality counting dilemma, leveraging a federated learning framework.
This approach involves a combination of local differentially private data encoding and privacy-aware federated k-means clustering strategy, ensuring that cardinality counting occurs across distinct datasets without necessitating data amalgamation.


pros:
1. This paper proposes a pratical method.
2. This paper proposes a privacy-preserving method for the cardinality counting dilemma, which is crucial in sectors like healthcare, marketing, cybersecurity, and web analytics.
3. This paper tests the method on both real-world and simulated datasets, which indicates the robustness of the proposed method.
4. This paper conducts detailed experiments.


cons:
1. More related works are needed to highlight the novelty of the proposed method.
2. Although Algorithm 2 summarizes the privacy-preserving federated K-means clustering algorithm for cardinality counting, more details on the steps of the algorithm should be provided in Section 3.4.
3. Federated learning can be resource-intensive, especially with privacy-preserving techniques like differentially private data encoding. This may pose challenges in terms of computational costs and efficiency. Is it possible to provide potential solutions to solve the above challenges?

**Questions:**

see cons.

**Reviewer Confidence:**

3: The reviewer is confident but not certain that the evaluation is correct

**Scope:**

4: The work is relevant to the Web and to the track, and is of broad interest to the community

---

### Official Review · Reviewer_aqda · 2023-11-23

**Novelty:** 4
**Technical Quality:** 4

**Review:**

This paper presents a privacy-aware approach to ensure cardinality counting for distinct datasets without amalgamation. The idea is to employ a locally differentially private data encoding followed by a privacy-aware K-means clustering.

Pros:
- Interesting and practical use case.
- Good writing overall.

Cons:
- Missing related literature and explanation for the design choice.
- Evaluation is not thorough.

Overall, I think this paper addresses an interesting problem in privacy-aware collaborations on sensitive datasets.

The design choice is unclear. The paper does not discuss existing literature to convince why the proposed design is effective. For example, Bloom filter and local DP encoding are known methods but this design choice was discussed too briefly. This missing related work (https://dl.acm.org/doi/10.1145/3372224.3419188) proposed a pretty similar design idea for submodel learning.

The paper only evaluates the counting accuracy. It is important to also discuss the overheads (e.g., computation, communication) and potential limitations in larger-scale data analysis collaborations.

**Questions:**

1. What are the existing practices other than Bloom filter and local DP?
2. What are the overheads or limitations?

**Reviewer Confidence:**

2: The reviewer is willing to defend the evaluation, but it is likely that the reviewer did not understand parts of the paper

**Scope:**

3: The work is somewhat relevant to the Web and to the track, and is of narrow interest to a sub-community

---

### Official Review · Reviewer_9oJj · 2023-11-25

**Novelty:** 3
**Technical Quality:** 4

**Review:**

This work proposes a privacy-preserving, FL-based solution for cardinality counting of datasets. The approach provides privacy guarantees by first applying a bloom filter encoding and then local DP on the client side. The author evaluated their approach using real and synthetic datasets.

Strengths of this work:
- Well-written and easy-to-follow manuscript
- Work is very well motivated.
- I appreciated all the background information, which makes the paper easy to read and understand

Weaknesses:
- My main complaint is the contribution of this work. The solution looks very trivial (Bloom filter encoding > PD > Local K-means > global K-means).
- The authors mention: "the first federated cardinality counting framework that allows cardinality counting to occur across distinct datasets". Why FL is important in your case? What are the limitations that FL solves?
- Limited related work (section is mixed with Introduction). Are there any related work? You mention "the existing methods do not fully consider privacy during the counting process" and refer to a supporting citation, but what are these works?
- Used baseline is a central clustering with local DL. I appreciate the effort, but would expect to also compare with other existing approaches.
- Possible missing related work: "Learning with Privacy at Scale", Differential Privacy Team, Apple

**Questions:**

- What are the available related works and how do they compare to your approach?

**Ethics Review Description:**

Datasets used are public

**Reviewer Confidence:**

3: The reviewer is confident but not certain that the evaluation is correct

**Scope:**

4: The work is relevant to the Web and to the track, and is of broad interest to the community

---

### Official Review · Reviewer_ouiC · 2023-11-28

**Novelty:** 5
**Technical Quality:** 5

**Review:**

# Summary

In this paper, the authors propose a method for cardinality counting within the federated learning framework. This approach enables cardinality counting in a privacy-preserving manner, without revealing the actual data.

# Strength
Good logic flow and writing.

# Weakness

The paper appears to overlook a few significant contributions in the field of privacy-preserving cardinality estimation. Despite the emphasis on the importance of this problem, there seems to be a lack of reference to some recent notable works on the topic.

- For instance, the paper by Wright et al., entitled "Privacy-Preserving Secure Cardinality and Frequency Estimation" (https://storage.googleapis.com/pub-tools-public-publication-data/pdf/3e44af84a8404c28aaebff347a4bd5e305a62eda.pdf), introduces advanced methods for cardinality and frequency estimation by combining aspects of HyperLogLog (HLL) and Bloom filters. This work is particularly relevant as it presents a scalable secure multi-party computation protocol that is crucial for the topic at hand.

- Another significant contribution is the NeurIPS 2020 paper, "The Flajolet-Martin Sketch Itself Preserves Differential Privacy: Private Counting with Minimal Space" (https://proceedings.neurips.cc/paper/2020/file/e3019767b1b23f82883c9850356b71d6-Paper.pdf). This paper discusses privacy preservation using Flajolet-Martin Sketch, which is a key technique in the realm of cardinality estimation.

Considering the relevance and impact of these works to the problem under discussion, it would greatly benefit the paper to include these in the discussion of related works. This could provide a more comprehensive background for the study, and further strengthen the positioning and novelty of the current work.

**Questions:**

Why the related works are not mentioned & evaluated?

**Ethics Review Description:**

No ethical issue

**Reviewer Confidence:**

2: The reviewer is willing to defend the evaluation, but it is likely that the reviewer did not understand parts of the paper

**Scope:**

4: The work is relevant to the Web and to the track, and is of broad interest to the community

---

### Decision · Program_Chairs · 2024-01-22

**Decision:**

Accept

**Comment:**

This paper propose a privacy-preserving federated clustering method for cardinality counting. The approach provides privacy guarantees by first applying a bloom filter encoding and then local DP on the client side, followed by privacy-aware K-means clustering.

 Most reviewers agree the paper is well-written and well-motivated. The idea, although not very novel, is technically sound and practical. The evaluation is extensive and results convincing.

 Suggestion: Please add more references to related work.